# PDE-GCN: Novel Architectures for Graph Neural Networks Motivated by Partial Differential Equations

**Moshe Eliasof**
Department of Computer Science
Ben-Gurion University of the Negev
Beer-Sheva, Israel
eliasof@post.bgu.ac.il

**Eldad Haber**
Department of Earth, Ocean and Atmospheric Sciences
University of British Columbia
Vancouver, Canada
ehaber@eoas.ubc.ca

**Eran Treister**
Department of Computer Science
Ben-Gurion University of the Negev
Beer-Sheva, Israel
erant@cs.bgu.ac.il

## Abstract

Graph neural networks are increasingly becoming the go-to approach in various fields such as computer vision, computational biology and chemistry, where data are naturally explained by graphs. However, unlike traditional convolutional neural networks, deep graph networks do not necessarily yield better performance than shallow graph networks. This behavior usually stems from the over-smoothing phenomenon. In this work, we propose a family of architectures to control this behavior by design. Our networks are motivated by numerical methods for solving Partial Differential Equations (PDEs) on manifolds, and as such, their behavior can be explained by similar analysis. Moreover, as we demonstrate using an extensive set of experiments, our PDE-motivated networks can generalize and be effective for various types of problems from different fields. Our architectures obtain better or on par with the current state-of-the-art results for problems that are typically approached using different architectures.

## 1 Introduction

In recent years, Graph Convolutional Networks (GCNs) [1, 2, 3] have drawn the attention of researchers and practitioners in a variety of domains and applications, ranging from computer vision and graphics [4, 5, 6, 7, 8] to computational biology [9, 10, 11], recommendation systems [12] and social network analysis [13, 14]. However, GCNs still suffer from two main problems. First, they are usually *shallow* as opposed to the concept of deep convolutional neural networks (CNNs) [15, 16] due to the *over-smoothing* phenomenon [17, 18, 19], where the node feature vectors become almost identical, such that they are indistinguishable, which yields non-optimal performance. Furthermore, GCNs are typically customized to a specific domain and application. That is, as we demonstrate in Sec. 4.1, a successful point-cloud classification network [5] can perform poorly on a citation graph node-classification problem [19], and vice-versa. Furthermore, because many GCNs lack theoretical guarantees, it is difficult to reason about their success on one problem and lack on others. These observations motivate us to develop a profound understanding of graph networks and their dynamics.

To this end, we suggest a novel, universal approach to the design of GCN architectures. Our inspiration stems from the similarities and equivalence between Partial Differential Equations (PDEs) and deep networks explored in [20, 21, 22]. Furthermore, as GCNs can be thought of as a generalization of

35th Conference on Neural Information Processing Systems (NeurIPS 2021).

CNNs, and a standard convolution can be represented as a combination of differential operators on a structured grid [22], we adopt this interpretation to explore versions of GCNs as PDEs on graphs or manifolds. We therefore call our network architectures *PDE-GCN*, and demonstrate that our approach is general with respect to the given task. That is, our architectures behave similarly for different problems, and their performance is on par or better than other domain-specific GCNs. Furthermore, our family of architectures are backed by theoretical guarantees that allow us to explain the behavior of the GCNs in some of the results that we present. To be more specific, our contribution is as follows:

- We introduce and implement general graph convolution operators, based on graph gradient and divergence. This abstraction of the spatial operation on the graph leads to a more general and flexible approach for architecture design.
- We propose treating a variety of graph related problems as discretized PDEs, and formulate the dynamics that match different problems such as node-classification and dense shape-correspondence. This is in direct effort to propose a family of networks that can solve multiple problems, instead of GCNs which are tailored for a specific application.
- Our method allows constructing a deep GCN without over-smoothing, with theoretical guarantees.
- We validate our method by conducting numerous experiments on multiple datasets and applications, achieving significantly better or similar accuracy compared to state-of-the-art models.

## 2   Related work

**Graph Convolutional Networks:**   GCNs are typically divided into spectral [1, 3, 2] and spatial [23, 24, 25, 4] categories. Most of those can be implemented using the Message-Passing Neural Network paradigm [25], where each node aggregates features (messages) from its neighbors, according to some scheme. The works [3, 2] use polynomials of the graph Laplacian to parameterize the convolution operator. DGCNN [5] constructs a k-nearest-neighbors graph from point-clouds and dynamically updates it. MoNet [4] learns a Gaussian mixture model to weight the edges of meshes for shape analysis tasks. Works like GraphSAGE [26] and GAT [27] propose methods for inductive and transductive learning on non-geometrical graphs.

Several of the methods above suffer from over-smoothing [17, 19], leading to undesired node features similarity for deep networks. To overcome this problem, some approaches rely on imposing regularization and augmentation. For example, PairNorm [17] introduces a novel normalization layer, and DropEdge [28] randomly removes edges to decrease the degree of nodes. Other methods prevent over-smoothing by dedicated construction. For instance, JKNet [29] combines all intermediate representations at each stage of the network. APPNP [30] replaces learnt convolutional layers with a pre-defined kernel based on PageRank, yielding a shallow network which preserves locality, making it robust to over-smoothing. GCNII [19] proposes to add an identity residual where the initial features of the networks are added to the features of the $l$-th layer, scaled by some coefficient.

Another approach is to construct a network that inherently does not over-smooth, as we suggest in this work. Our network is based on discretized PDEs, hence we are able to motivate our choices by well studied theory and numerical experiments [31]. On a similar note, the recent DiffGCN [8] also makes use of discretized operators to approximate the graph gradient and Laplacian. However, DiffGCN is specifically tailored for geometric tasks since it projects its components on the $x, y, z$ axes, and is applied using a ResNet [16] (diffusion) structure only. The recent GRAND [32] applies attention mechanism with diffusive dynamics, using several integration schemes. Here we propose a network for both geometric and non-geometric tasks, and also utilize both the diffusion or hyperbolic layer dynamics, including a mixture of the two.

**PDEs and CNNs:**   In a recent series of works, the connection between PDEs and CNNs was studied [20, 21, 22, 33, 34, 35, 36, 37]. It was shown that it is possible to treat a deep neural network as a dynamical system driven by some PDE, where each convolution layer is considered a time step of a discretized PDE. The connection between PDEs and CNNs was also used to reduce the computational burden [38]. Besides the interpretation of CNNs as a PDE solver, it was shown that it is also possible to learn a symbolic representation of PDEs in a task-and data driven approach in [39, 40]. In the context of GCNs and PDEs, it was recently shown [41, 42] that GCNs can be utilized to enhance the solution of PDEs for problems like fluid flow dynamics. However, in this work, we harness PDE concepts to design and construct GCNs for a variety of applications.

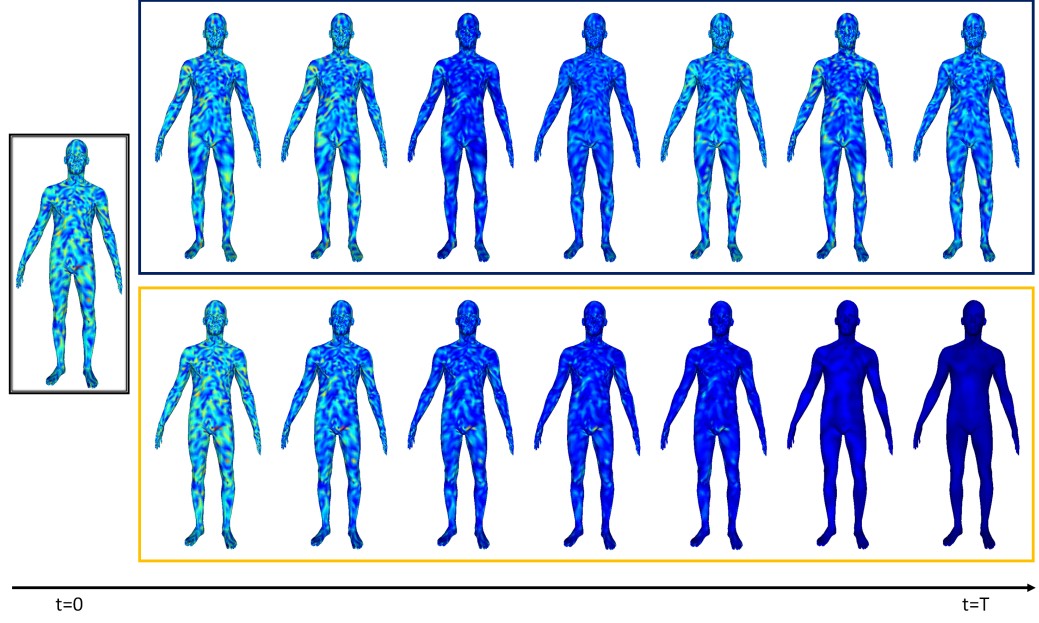

t=0                                                                                    t=T

Figure 1: Feature evolvement on an input mesh (left). Propagation in time is from left to right. Hyperbolic and diffusion equation dynamics are on the top and bottom row, respectively. While a diffusive graph network (similar to most common GCNs that rely on ResNets) smooths the information on the manifold, the hyperbolic network yields a non-uniform field.

## 3 Methods

### 3.1 Partial Differential Equations on manifolds

We show now that GCNs can be viewed as discretizations of PDEs on manifolds, similarly to CNNs that are viewed as discretized PDEs on regular grids [22]. Consider a general manifold $\mathcal{M}$ where a vector function $f$ resides (also dubbed the features function), along with its continuous differential operators such as the gradient $\nabla$, divergence $\nabla\cdot$ and the Laplacian $\Delta$ that reside on the manifold $\mathcal{M}$.

Given these differential operators, one can model different processes on $\mathcal{M}$. In particular, we consider two PDEs – the non-linear diffusion and the non-linear hyperbolic equations

$$f_t = \nabla \cdot K^*\sigma(K\nabla f), \quad f(t=0) = f^0, \quad t \in [0, T], \tag{1}$$

$$f_{tt} = \nabla \cdot K^*\sigma(K\nabla f), \quad f(t=0) = f^0, \quad f_t(t=0) = 0, \quad t \in [0, T], \tag{2}$$

respectively, equipped with appropriate boundary conditions. Here $K$ is a coefficient matrix that can change in time and represents the propagation over the manifold $\mathcal{M}$, $K^*$ is its conjugate transpose and $\sigma(\cdot)$ is a non-linear activation function. Eq. (1)-(2) define a non-linear operator that takes initial feature vectors $f^0$ at time 0 and propagates them to time $T$, yielding $f^T$ where they can be used for different tasks. We now provide two theorems that characterize the behavior of Eq. (1)-(2), based on ideas from [22][1].

**Theorem 1.** *If the activation function $\sigma(\cdot)$ is monotonically non-decreasing and sign-preserving, then the forward propagation through the diffusive PDE in* (1) *for $t \in [0, \infty)$ yields a non-increasing feature norm, that is,*

$$\frac{\partial}{\partial t}\|f\|^2 \le 0.$$

**Theorem 2.** *Assume that the activation function $\sigma(\cdot)$ is monotonically non-decreasing, sign-preserving and satisfies $|\sigma(x)| \le |x|$, and define energy*

$$\mathcal{E}_{net} = \|f_t\|^2 + (K\nabla f, \sigma(K\nabla f)),$$

---

[1]See proofs in Appendix A.

*then the forward propagation through the hyperbolic PDE in (2) satisfies $\mathcal{E}_{net} \leq c_K$, where $c_K$ is a constant that depends on $K$ but independent of time.*

The outcome of those theorems is that the dynamics described in Eq. (1) is smoothing, while the one in Eq. (2) is bounded by a conserving mapping. An illustration of this behavior is presented in Fig. 1.

In the physical world, diffusion and hyperbolic equations are used for different applications. Similarly, many computational models for image segmentation [43], denoising [44] and deblurring are based on anisotropic diffusion which are similar to the model in Eq. (1). On the other hand, applications that require conservation such as volume/distance preservation as in the dense shape correspondence task [45] and protein folding [11], are typically better treated using a hyperbolic equation as in Eq. (2). Those insights motivate us to construct two types of layers according to Eq. (1)-(2) using discretized differential operators on graphs.

### 3.2 Discretized differential operators on graphs

The models (1) and (2) reside in a continuous manifold $\mathcal{M}$, on which a continuous function vector $f$ is defined. A graph can be thought of as a discretization of that manifold to a finite space. Assume we are given an undirected graph $\mathcal{G} = (\mathcal{V}, \mathcal{E})$ where $\mathcal{V} \in \mathcal{M}$ is the set of $n$ vertices of the graph and $\mathcal{E}$ is the set of $m$ edges of the graph. Let us denote by $\mathbf{f}_i \in \mathbb{R}^c$ the value of the discrete version of $f$, on the $i$-th node of $\mathcal{G}$. $c$ is the number of channels, which is the width of the neural network. We define $\mathbf{G}$, the discrete gradient operator on the graph, also known as the incidence matrix, as follows:

$$(\mathbf{G}\mathbf{f})_{ij} = \mathbf{W}_{ij}(\mathbf{f}_i - \mathbf{f}_j), \tag{3}$$

where nodes $i$ and $j$ are connected via the $(i, j)$-th edge, $\mathbf{W}_{ij}$ is an edge weight matrix which can be learnt, and $\mathbf{f}_i$ and $\mathbf{f}_j$ are the features on the $i$-th and $j$-th nodes, respectively. The gradient operator can be thought of as a weighted directional derivative of the function $f$ in the direction defined by the nodes $i$ and $j$. Furthermore, if we choose the scaled identity matrix $\mathbf{W}_{ij} = d_{ij}^{-1}\mathbf{I}$, where $d_{ij}$ is the distance between the two nodes, then the discrete gradient is a second order approximation to the true gradient of the function $f$ on the edges of the graph. In this work, we use $\mathbf{W}_{ij} = \gamma_{ij}\mathbf{I}$, where the scale $\gamma_{ij}$ is the geometric mean of the degree of the nodes $i, j$. Note, that the gradient operator is a mapping from the *vertex* space to the *edge* space.

Given the gradient matrix, it is possible to define the divergence matrix [46], which is an approach that is used extensively in mimetic discretizations of PDEs. To this end, we define the inner product between an edge feature vector $\mathbf{q}$ and the gradient of a node feature vector $\mathbf{f}$ as

$$(\mathbf{q}, \mathbf{G}\mathbf{f}) = \mathbf{q}^\top \mathbf{G}\mathbf{f} = \mathbf{f}^\top \mathbf{G}^\top \mathbf{q}. \tag{4}$$

The divergence is naturally defined as the operator that maps edge operator $\mathbf{q}$ to the node space, that is $\nabla \cdot \approx -\mathbf{G}^\top$. As usual, the graph Laplacian operator can be obtained by taking the divergence of the gradient. In graph theory it is defined as a positive matrix that is, $\Delta \approx \mathbf{G}^\top \mathbf{G}$ [2].

We also define the weighted line integral over an edge. Similarly to Eq. (3)-(4), we define

$$(\mathbf{A}\mathbf{f})_{ij} = \frac{1}{2}\mathbf{W}_{ij}(\mathbf{f}_i + \mathbf{f}_j), \quad (\mathbf{q}, \mathbf{A}\mathbf{f}) = \mathbf{q}^\top \mathbf{A}\mathbf{f} = \mathbf{f}^\top \mathbf{A}^\top \mathbf{q}. \tag{5}$$

The operator $\mathbf{A}$ approximates the mass operator on the edges. The right equation in Eq. (5) suggests that an appropriate averaging operator for edge features is the transpose of the nodal edge average.

The advantage of defining such operators is that we are able to design networks with architectures that mimic the continuous operators (1) and (2) on the discrete level, as we show in the next section.

### 3.3 PDE-GCN: Graph Convolutional Networks by Partial Differential Equations

In order to use the computational models in Eq. (1)-(2), we form their discrete versions:

$$\mathbf{f}^{(l+1)} = \mathbf{f}^{(l)} - h\mathbf{G}^\top \mathbf{K}_l^\top \sigma(\mathbf{K}_l \mathbf{G}\mathbf{f}^{(l)}), \tag{6}$$

$$\mathbf{f}^{(l+1)} = 2\mathbf{f}^{(l)} - \mathbf{f}^{(l-1)} - h^2\mathbf{G}^\top \mathbf{K}_l^\top \sigma(\mathbf{K}_l \mathbf{G}\mathbf{f}^{(l)}). \tag{7}$$

---

[2]In the field of numerical PDEs, the Laplacian is defined as $-\mathbf{G}^\top \mathbf{G}$, but the combinatorial Laplacian is defined as $\mathbf{G}^\top \mathbf{G}$.

Here, in Eq. (6) we use the forward Euler to discretize Eq. (1), and in Eq. (7) we discretize the second order time derivative in Eq. (2), using the leapfrog method. In both cases, $\mathbf{f}^{(l)}$ are the node features and $\mathbf{K}_l$ is a $1 \times 1$ trainable convolution of the $l$-th layer. The similarity between (6) and ResNet is well documented in the context of CNNs [22]. The hyper-parameter $h$ is the step-size, and it is chosen such that the stability of the discretization is kept. We use $\sigma = \tanh$ for the activation function as it yields slightly better results in our experiments, although other functions such as ReLU can also be used, as reported in Sec. 4.7. Each of Eq. (6)-(7) defines a PDE-GCN block. We denote the former (diffusive equation) by PDE-GCN$_\text{D}$ and the latter (hyperbolic equation) by PDE-GCN$_\text{H}$.

To complete the description of our network, a few more details are required, as follows:

**The (opening) embedding layer.** The input vertex features $\mathbf{u}_\mathcal{V}$ are fed through an embedding ($1 \times 1$ convolution) layer $\mathbf{K}_o$ to obtain the initial features $\mathbf{f}_0$ of our PDE-GCN network: $\mathbf{f}_0 = \mathbf{K}_o \mathbf{u}_\mathcal{V}$. In cases where input edge attributes (features) $\mathbf{u}_\mathcal{E}$ are available (as in the experiment in Sec. 4.6), we transform them to the vertex space by taking their divergence and average, and concatenate them to the input of the embedding layer $\mathbf{K}_o$ as follows: $\mathbf{f}_0 = \mathbf{K}_o(\mathbf{u}_\mathcal{V} \oplus \mathbf{A}^\top \mathbf{u}_\mathcal{E} \oplus \mathbf{G}^\top \mathbf{u}_\mathcal{E})$.

**The (closing) embedding layer.** Given the final vertex features $\mathbf{f}^{(L)}$ of our PDE-GCN with $L$ layers, we obtain the output of the network by performing: $\mathbf{u}_{out} = \mathbf{K}_c \mathbf{f}^{(L)}$. Here $\mathbf{K}_c$ is a $1 \times 1$ convolution layer mapping the hidden feature space to the output shape.

**Initialization.** The $1 \times 1$ convolutions $\mathbf{K}_l$ in Eq. (6)-(7) are initialized as identity. This way, the network begins from a diffusion/hyperbolic equation, which serves as a prior and guides the network to initially behave like classical methods [43, 45] and to further improve during training.

**The choice of dynamics.** For some applications, anisotropic diffusion is appropriate, while for others conservation is more important. However, in some applications this may not be clear. To this end, it is possible to combine Eq. (1)-(2) to obtain the continuous process

$$\alpha f_{tt} + (1 - \alpha)f_t = \nabla \cdot K^* \sigma(K \nabla f), \quad f(t=0) = f^0, \quad f_t(t=0) = 0 \quad t \in [0, T], \quad (8)$$

where $\alpha = sigmoid(\beta)$, meaning $0 \leq \alpha \leq 1$, and $\beta$ is a single trainable parameter. The discretization of this PDE leads to the following network dynamics:

$$\alpha(\mathbf{f}^{(l+1)} - 2\mathbf{f}^{(l)} + \mathbf{f}^{(l-1)}) + h(1 - \alpha)(\mathbf{f}^{(l+1)} - \mathbf{f}^{(l)}) = -h^2 \mathbf{G}^\top \mathbf{K}_l^\top \sigma(\mathbf{K}_l \mathbf{G} \mathbf{f}^{(l)}), \quad (9)$$

where $\mathbf{f}^{(l+1)}$ is updated by the known $\mathbf{f}^{(l)}$ and $\mathbf{f}^{(l-1)}$. We denote a layer that is governed by Eq. (9) by PDE-GCN$_\text{M}$ (where M stands for mixture). Note, that it is also possible to learn a combination coefficient $\alpha_i$ per layer, although we did not read a benefit from such scheme.

As we show in our numerical experiments, learning $\alpha$ yields results that are consistent with our understanding of the problem, that is, graph node-classification gravitates towards no second order derivatives while applications that require conservation gravitate towards the hyperbolic equation.

## 4 Experiments

In this section we demonstrate our approach on various problems from different fields and applications ranging from 3D shape-classification [47] to protein-protein interaction [26] and node-classification [48]. The experiments also vary in their output type. Node classification is similar to segmentation problems that are typically solved by anisotropic diffusion while the dense shape correspondence problem is conservative and therefore can be thought of as a problem that does not require smoothing.

In all experiments, we use the suitable PDE-GCN (D, H or M) block as described in Sec. 3, with various depths (number of layers) and widths (number of channels), as well as the appropriate final convolution steps, depending on the task at hand. A detailed description of the architectures used in our experiments is given in Appendix B. We use the Adam [49] optimizer in all experiments, and perform grid search over the hyper-parameters of our network. The selected hyper-parameters are reported in Appendix C. Our objective function in all experiments is the cross-entropy loss, besides inductive learning on PPI where we use the binary cross-entropy loss. Our code is implemented using PyTorch [50], trained on an Nvidia Titan RTX GPU.

We show that for all the considered tasks and datasets, our method is either remarkably better or on par with state-of-the-art models.

Table 1: Generalization of GCNs to different domains and applications. $(L)$ denotes $L$ layers.

| Method | Dataset | Accuracy (%) |
|---|---|---|
| DGCNN (4) [5] | ModelNet-10 | 92.8 |
| DGCNN (2) / (4) | Cora | 34.9 / 25.2 |
| DGCNN + Diffusion (2) / (4) | Cora | 71.0 / 66.1 |
| GCNII (4) [19] | ModelNet-10 | 65.4 |
| | Cora | 82.6 |
| PDE-GCN$_D$(4) (Ours) | ModelNet-10 | 92.2 |
| | Cora | 83.6 |

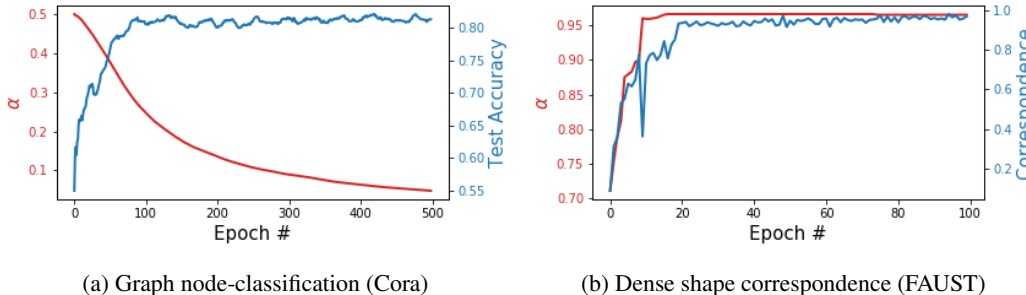

(a) Graph node-classification (Cora)  (b) Dense shape correspondence (FAUST)

Figure 2: Learnt mixture of the hyperbolic equation $\alpha$, to the dynamics of the network. The diffusion equation contribution is $1 - \alpha$.

## 4.1 Generalization of GCNs to different applications

To gain deeper understanding about the effectiveness and generalization of various GCN methods to different tasks, we start by picking two datasets - ModelNet-10 [47] for 3D shape-classification, and Cora [48] for semi-supervised node classification. Those datasets are not only different in terms of application (global versus local classification), but also stem from different domains. While in ModelNet-10 the data has geometrical meaning, the data in Cora has no obvious geometrical interpretation. Therefore, we suggest that success in both applications should be obtained from a generalizable GCN. We compare our PDE-GCN$_D$ with two recent and popular networks - DGCNN [5] and GCNII [19]. For ModelNet-10 shape-classification, we randomly sample 1,024 points from each shape to form a point cloud, and connect its points using k-nearest-neighbors (k-nn) algorithm with $k = 10$ to obtain a graph and follow the training scheme of [5]. On Cora, we follow the same procedure as in [19]. We evaluate all models with 4 layers, as well as 2 layers for DGCNN on Cora.

Our results, reported in Tab. 1 suggest that while each of the considered methods obtains high accuracy on the dataset it originally was tested on (Cora for GCNII and ModelNet-10 for DGCNN), obtaining a similar measure of success on a different dataset was not possible when using the very same networks. Additionally, while our attempts to add a diffusion-equation dynamics to DGCNN (i.e., updating features as in Eq. (6)) showed an increase in performance – a large gap to state-of-the-art model still exists. On top of that, we also see that DGCNN suffers from over-smoothing, as its accuracy significantly decreases when adding more layers. Last but not least, we observe that our PDE-GCN$_D$ obtains high accuracy on both datasets, similar or better than state-of-the-art models.

## 4.2 Learning PDE network dynamics

In this experiment, we delve on the ability to *learn* the appropriate PDE that better models a given problem. To this end, we use the mixture model from Eq. (9) so that the resulting PDE is a combination of the diffusion and hyperbolic dynamics. We use a 8 layer mixed PDE-GCN, starting with $\alpha = 0.5$, such that it is balanced between a PDE-GCN$_D$ and PDE-GCN$_H$. By learning the parameter $\alpha$ in (9), we allow to choose a mixed PDE between a purely conservative network and a

diffusive one. We consider two problems: semi-supervised node classification on Cora, and dense shape correspondence on FAUST [51].

Our results, reported in Fig. 2 suggest that just as in classical works [43, 52], problems like node-classification obtain better performance with an anisotropic diffusion like in Eq. (6), and for problems involving dense-correspondences like in [45, 11] that tend to conserve the energy of the underlying problem, a hyperbolic equation type of PDE as in Eq. (7) is more appropriate.

Table 2: Statistics of datasets used in our semi-and fully supervised node-classification experiments.

| Dataset | Cora | CiteSeer | PubMed | Chameleon | Cornell | Texas | Wisconsin | PPI |
|---|---|---|---|---|---|---|---|---|
| Classes | 7 | 6 | 3 | 5 | 5 | 5 | 5 | 121 |
| Nodes | 2,708 | 3,327 | 19,717 | 2,277 | 183 | 183 | 251 | 56,944 |
| Edges | 5,429 | 4,732 | 44,338 | 36,101 | 295 | 309 | 499 | 818,716 |
| Features | 1,433 | 3,703 | 500 | 2,325 | 1,703 | 1,703 | 1,703 | 50 |

### 4.3 Semi-supervised node classification

In this set of experiments we use three datasets – Cora, Citeseer and PubMed [48]. For all datasets we use the standard training/validation/testing split as in [53], with 20 nodes per class for training, 500 validation nodes and 1,000 testing nodes and follow the training scheme of [19]. The statistics of the datasets are reported in Tab. 2. We compare our results using PDE-GCN$_D$ with recent and popular models like GCN [3], GAT [27], APPNP [30], JKNet [29] and DropEdge [28]. We note that our network does not over-smooth, as an increase in the number of layers does not cause performance degradation. For example, on CiteSeer, we obtain 75.6% accuracy with 32 layers, compared to 74.6% with two layers[3]. Overall, our results in Tab. 3 show that our PDE-GCN$_D$ achieves similar or better accuracy than the considered methods.

### 4.4 Fully-supervised node classification

We follow [54] and use 7 datasets: Cora, CiteSeer, PubMed, Chameleon, Cornell, Texas and Wisconsin. We also use the same train/validation/test splits of $60\%, 20\%, 20\%$, respectively. In addition, we report the average performance over 10 random splits from [54]. We fix the number of channels to 64 and perform grid search to determine the hyper-parameters parameters, which are reported in Appendix C. We compare our network with GCN, GAT, three variants of Geom-GCN [54], APPNP, JKNet, Incep and GCNII in Tab. 4. Our experiments read either similar or better than the state-of-the-art on Cora, CiteSeer and PubMed datasets. On Chameleon [55], Cornell, Texas and Wisconsin datasets, we improve state-of-the-art accuracy by a significant margin. For example, we obtain $93.24\%$ accuracy on Texas with our PDE-GCN$_M$, compared to $77.84\%$ with GCNII*. Similar improvements hold for Cornell and Wisconsin datasets. The common factor for these datasets is their small size, as depicted from Tab. 2. We argue that the success of our network stems from its capability of apriori extracting features from graphs, due to its utilization of discretized differential operators and PDE guided construction. On Chameleon [55], using PDE-GCN$_M$, we improve the current state-of-the-art accuracy of GCNII* from $62.48\%$ to $66.01\%$. Also, we note that unlike in the semi-supervised case, where some of the labels are missing, here it is possible to obtain meaningful results with the hyperbolic equation based PDE-GCN$_H$ as we do not have unknown nodes in the fully-supervised case, which would be otherwise preserved using the hyperbolic equation dynamics.

### 4.5 Inductive Learning

We follow [19] and employ the PPI dataset [26] for the inductive learning task. We use a 8 layer PDE-GCN$_D$ network, without dropout or weight-decay, and a learning rate of 0.001. We compare our results with methods like GraphSAGE, GAT, JKNet, GeniePath, GCNII and others. As reported in Tab. 5, our PDE-GCN$_D$ achieves 99.07 Micro-averaged F1 score, superior to methods like GAT, JKNet and GeniePath, also close to state-of-the-art GCNII* with a score of 99.58.

---

[3]Note that this result is also a new state-of-the-art accuracy.

Table 3: Semi-supervised node classification accuracy (%). – indicates not available results.

| Dataset | Method | Layers | | | | | |
|---|---|---|---|---|---|---|---|
| | | 2 | 4 | 8 | 16 | 32 | 64 |
| Cora | GCN [3] | **81.1** | 80.4 | 69.5 | 64.9 | 60.3 | 28.7 |
| | GCN (Drop) [28] | **82.8** | 82.0 | 75.8 | 75.7 | 62.5 | 49.5 |
| | JKNet [29] | – | 80.2 | 80.7 | 80.2 | **81.1** | 71.5 |
| | JKNet (Drop) [28] | – | **83.3** | 82.6 | 83.0 | 82.5 | 83.2 |
| | Incep [28] | – | 77.6 | 76.5 | 81.7 | **81.7** | 80.0 |
| | Incep (Drop) [28] | – | 82.9 | 82.5 | 83.1 | 83.1 | **83.5** |
| | GCNII [19] | 82.2 | 82.6 | 84.2 | 84.6 | 85.4 | **85.5** |
| | GCNII* [19] | 80.2 | 82.3 | 82.8 | 83.5 | 84.9 | **85.3** |
| | PDE-GCN$_D$ (Ours) | 82.0 | 83.6 | 84.0 | 84.2 | 84.3 | **84.3** |
| Citeseer | GCN [3] | **70.8** | 67.6 | 30.2 | 18.3 | 25.0 | 20.0 |
| | GCN (Drop) [28] | **72.3** | 70.6 | 61.4 | 57.2 | 41.6 | 34.4 |
| | JKNet [29] | – | 68.7 | 67.7 | **69.8** | 68.2 | 63.4 |
| | JKNet (Drop) [28] | – | 72.6 | 71.8 | **72.6** | 70.8 | 72.2 |
| | Incep [28] | – | 69.3 | 68.4 | **70.2** | 68.0 | 67.5 |
| | Incep (Drop) [28] | – | **72.7** | 71.4 | 72.5 | 72.6 | 71.0 |
| | GCNII [19] | 68.2 | 68.8 | 70.6 | 72.9 | **73.4** | 73.4 |
| | GCNII* [19] | 66.1 | 66.7 | 70.6 | 72.0 | **73.2** | 73.1 |
| | PDE-GCN$_D$ (Ours) | 74.6 | 75.0 | 75.2 | 75.5 | **75.6** | 75.5 |
| Pubmed | GCN [3] | **79.0** | 76.5 | 61.2 | 40.9 | 22.4 | 35.3 |
| | GCN (Drop) [28] | **79.6** | 79.4 | 78.1 | 78.5 | 77.0 | 61.5 |
| | JKNet [29] | – | 78.0 | **78.1** | 72.6 | 72.4 | 74.5 |
| | JKNet (Drop) [28] | – | 78.7 | 78.7 | **79.7** | 79.2 | 78.9 |
| | Incep [28] | – | 77.7 | **77.9** | 74.9 | – | – |
| | Incep (Drop) [28] | – | **79.5** | 78.6 | 79.0 | – | – |
| | GCNII [19] | 78.2 | 78.8 | 79.3 | **80.2** | 79.8 | 79.7 |
| | GCNII* [19] | 77.7 | 78.2 | 78.8 | **80.3** | 79.8 | 80.1 |
| | PDE-GCN$_D$ (Ours) | 79.3 | **80.6** | 80.1 | 80.4 | 80.2 | 80.3 |

## 4.6 Dense shape correspondence

Finding dense correspondences between shapes is a classical experiment for hyperbolic dynamics, as we are interested in modeling local motion dynamics. In essence, learning to find correspondences between shapes is similar to learning a transformation from one shape to the other. To this end, we use the FAUST dataset [51] containing 10 scanned human shapes in 10 different poses with 6,890 nodes each. We follow the train and test split from [4], where the first 80 subjects are used for training and the remaining 20 subjects for testing. Our metric is the correspondence percentage with zero geodesic error, i.e., the percentage of perfectly matched shapes from all our test cases. We follow the pre-processing and training scheme of [62] where we use Cartesian coordinates to describe distances between nodes, with initial features of a constant $\mathbf{1} \in \mathbb{R}^n$ where $n$ is the number of nodes. We use a 8 layer PDE-GCN$_D$ and PDE-GCN$_H$ variants, both with constant learning rate of 0.001 with no weight-decay or dropout, and compare to recent and popular methods like ACNN [60], MoNet [4], FMNet [61], and SplineCNN [62]. As expected from the discussion in Sec. 3.1, and reported in Tab. 6, the hyperbolic equation proves to be a better fit for this kind of problem. Also, our PDE-GCN$_H$ achieves a promising 99.9% correspondence rate with zero geodesic error, outperforming the rest of the considered methods.

## 4.7 Ablation study

Our method has two main dynamics – diffusion and hyperbolic. To verify our proposal in Sec. 3.1, we examine the performance of the hyperbolic PDE-GCN$_H$ on semi-supervised node-classification on Cora and CiteSeer datasets. Our results in Tab. 7 show that indeed for problems where we wish to obtain a piecewise-constant prediction, the diffusive formulation of our network, PDE-GCN$_D$, is

Table 4: Fully-supervised node classification accuracy (%). (L) indicates a $L$ layers network.

| Method | Cora | Cite. | Pubm. | Cham. | Corn. | Texas | Wisc. |
|---|---|---|---|---|---|---|---|
| GCN [3] | 85.77 | 73.68 | 88.13 | 28.18 | 52.70 | 52.16 | 45.88 |
| GAT [27] | 86.37 | 74.32 | 87.62 | 42.93 | 54.32 | 58.38 | 49.41 |
| Geom-GCN-I [54] | 85.19 | 77.99 | 90.05 | 60.31 | 56.76 | 57.58 | 58.24 |
| Geom-GCN-P [54] | 84.93 | 75.14 | 88.09 | 60.90 | 60.81 | 67.57 | 64.12 |
| Geom-GCN-S [54] | 85.27 | 74.71 | 84.75 | 59.96 | 55.68 | 59.73 | 56.67 |
| APPNP [30] | 87.87 | 76.53 | 89.40 | 54.30 | 73.51 | 65.41 | 69.02 |
| JKNet [29] | 85.25 (16) | 75.85 (8) | 88.94 (64) | 60.07 (32) | 57.30 (4) | 56.49 (32) | 48.82 (8) |
| JKNet (Drop) [28] | 87.46 (16) | 75.96 (8) | 89.45 (64) | 62.08 (32) | 61.08 (4) | 57.30 (32) | 50.59 (8) |
| Incep (Drop) [28] | 86.86 (8) | 76.83 (8) | 89.18 (4) | 61.71 (8) | 61.62 (16) | 57.84 (8) | 50.20 (8) |
| GCNII [19] | 88.49 (64) | 77.08 (64) | 89.57 (64) | 60.61 (8) | 74.86 (16) | 69.46 (32) | 74.12 (16) |
| GCNII* | 88.01 (64) | 77.13 (64) | **90.30** (64) | 62.48 (8) | 76.49 (16) | 77.84 (32) | 81.57 (16) |
| PDE-GCN$_D$ (Ours) | 88.51 (16) | 78.36 (64) | 89.6 (64) | 64.12 (8) | 89.19 (2) | 90.81 (8) | 90.39 (8) |
| PDE-GCN$_H$ (Ours) | 87.71 (32) | 78.13 (16) | 89.16 (16) | 61.57 (64) | 89.45 (64) | 92.16 (64) | 91.37 (16) |
| PDE-GCN$_M$ (Ours) | **88.60** (16) | **78.48** (32) | 89.93 (16) | **66.01** (16) | **89.73** (64) | **93.24** (32) | **91.76** (16) |

Table 5: Protein-protein interaction (PPI). Results are reported in micro-averaged F1 score.

| Method | Micro-averaged F1 |
|---|---|
| GraphSAGE [26] | 61.20 |
| VR-GCN [56] | 97.80 |
| GaAN [57] | 98.71 |
| GAT [27] | 97.30 |
| JKNet [29] | 97.60 |
| GeniePath [58] | 98.50 |
| Cluster-GCN [59] | 99.36 |
| GCNII [19] | 99.54 |
| GCNII* [19] | **99.58** |
| PDE-GCN$_D$ (Ours) | 99.07 |
| PDE-GCN$_M$ (Ours) | 99.18 |

Table 6: Dense shape correspondence (%) with zero geodesic error

| Method | Faust |
|---|---|
| ACNN [60] | 63.8 |
| MoNet [4] | 89.1 |
| FMNet [61] | 98.2 |
| SplineCNN [62] | 99.2 |
| PDE-GCN$_D$ (Ours) | 64.2 |
| PDE-GCN$_H$ (Ours) | **99.9** |

more suitable. Furthermore, we study the importance of the positive-semi definiteness of our learnt operator as described in Eq. (6)-(7), by removing the $\mathbf{K}_l^T$ term from the dynamics equations. This yields a non-symmetric operator that does not guarantee positive-semi-definiteness. We note that the enforcement of the latter is important to obtain higher accuracy which is improved by up to $3\%$ with the introduction of a positive semi-definite operator. Also, we report on the use of $\sigma = ReLU$, from the discussion in Sec. 3.3, where we favor tanh as an activation function, and the ability of our PDE-GCN$_M$ to reproduce the results of PDE-GCN$_D$ in the case of semi-supervised learning.

# 5   Summary

In this paper we explored new architectures for graph neural networks. Our motivation stems from the similarities between graph networks and time dependent partial differential equations that are discretized on manifolds and graphs. By adopting an appropriate PDE, and embedding the finite graph in an infinite manifold, we are able to define networks that are either diffusive, conservative, or a combination of both.

Not all natural phenomena are solved using the same PDE and we should not expect that all graph problems should be solved by the same network dynamics. To this end we allow the data to choose which type of network is appropriate for the solution of the problem (diffusive or hyperbolic). Indeed, numerical experiments show that the network gravitates towards a hyperbolic one for problems where conservation is required, and towards a diffusive one when anisotropic diffusion is favorable.

Table 7: Ablation study of PDE-GCN accuracy (%) on semi-supervised node-classification.

| Method | Dataset | Layers | | | | | |
|---|---|---|---|---|---|---|---|
| | | 2 | 4 | 8 | 16 | 32 | 64 |
| PDE-GCN$_H$ | Cora | 79 | 79.3 | 78.0 | 78.0 | 77.8 | 77.5 |
| | CiteSeer | 70.9 | 71.7 | 72.1 | 72.3 | 72.5 | 72.4 |
| PDE-GCN$_D$ (non-symmetric) | Cora | 83.5 | 83.3 | 83.6 | 83.1 | 82.7 | 81 |
| | CiteSeer | 74.3 | 74.5 | 74.8 | 75.0 | 73.9 | 73.3 |
| PDE-GCN$_D$ ($\sigma = ReLU$) | Cora | 80.3 | 81.8 | 82.6 | 83.0 | 83.4 | 83.5 |
| | CiteSeer | 73.1 | 73.2 | 72.8 | 73.3 | 73.6 | 74.0 |
| PDE-GCN$_M$ | Cora | 82.0 | 83.4 | 83.9 | 84.2 | 84.3 | 84.5 |
| | CiteSeer | 74.2 | 74.8 | 75.3 | 75.4 | 75.4 | 75.8 |

Finally, we showed that the proposed networks can be made deep without over-smoothing and, can deliver the state-of-the-art performance or improve it for virtually every problem we worked with. In particular, our network dramatically improved the state-of-the-art for problems that are data-poor. We believe that for such problems the structure imposed by our dynamics and operators regularizes the network and therefore yields implicit regularization.

## Acknowledgments and Disclosure of Funding

The research reported in this paper was supported by grant no. 2018209 from the United States - Israel Binational Science Foundation (BSF), Jerusalem, Israel. ME is supported by Kreitman High-tech scholarship.

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
