# A Theorems and proofs

We repeat the theorems presented in Sec. 3 and provide their proofs below. The theorems hold for Neumann boundary conditions, which we use in our implementation—this is achieved by the construction of the differential operators. The proofs follow the ones presented in [22].

**Theorem 1.** *If the activation function $\sigma(\cdot)$ is monotonically non-decreasing and sign-preserving, then the forward propagation through the diffusive PDE in* (1) *for $t \in [0, \infty)$ yields a non-increasing feature norm, that is,*

$$\frac{\partial}{\partial t}\|f\|^2 \leq 0.$$

*Proof.* Let us examine the following inner product following Eq. (1):

$$(f, f_t) = (f, \nabla \cdot K^* \sigma(K\nabla f))$$

From integration by parts it holds that :

$$\frac{1}{2}\frac{\partial}{\partial t}\|f\|^2 = -(\nabla f, K^* \sigma(K\nabla f)) = -(K\nabla f, \sigma(K\nabla f)).$$

Plugging the definition of an inner product, together with the assumption that $\sigma$ is a sign-preserving function, it follows that:

$$sign(K\nabla f) = sign(\sigma(K\nabla f)).$$

Therefore, the following is non-positive:

$$\frac{1}{2}\frac{\partial}{\partial t}\|f\|^2 = -(K\nabla f, \sigma(K\nabla f)) \leq 0$$

Meaning

$$\frac{\partial}{\partial t}\|f\|^2 \leq 0.$$

$\square$

**Theorem 2.** *Assume that the activation function $\sigma(\cdot)$ is monotonically non-decreasing, sign-preserving and satisfies $|\sigma(x)| \leq |x|$, and define energy*

$$\mathcal{E}_{net} = \|f_t\|^2 + (K\nabla f, \sigma(K\nabla f)),$$

*then the forward propagation through the hyperbolic PDE in* (2) *satisfies $\mathcal{E}_{net} \leq c_K$, where $c_K$ is a constant that depends on $K$ and independent of time.*

*Proof.* Let us define the following energy:

$$\mathcal{E}_{lin} = \|f_t\|^2 + (K\nabla f, K\nabla f)$$

This energy is associated with the linear hyperbolic (wave-like) equation:

$$f_{tt} = \nabla \cdot K^* K\nabla f \quad f(t = 0) = f^0, \quad , \quad f_t(t = 0) = 0 \quad t \in [0, T].$$

Assuming $K$ is constant in time, we obtain:

$$\frac{1}{2}\partial_t \mathcal{E}_{lin} = (f_t, f_{tt} - \nabla \cdot K^* K\nabla f) = 0$$

This means that the energy $\mathcal{E}_{lin}$ is constant in time, i.e. there exists some $c_K$ such that $\mathcal{E}_{lin} = c_K$.

Also, given our assumption that $\sigma$ is sign-preserving and $|\sigma(x)| \leq |x|$ (i.e., it does not increase the norm of its input), we show that $\mathcal{E}_{net} \leq \mathcal{E}_{lin}$:

$$\mathcal{E}_{net} = \|f_t\|^2 + (K\nabla f, \sigma(K\nabla f))$$
$$\leq \|f_t\|^2 + (K\nabla f, K\nabla f) = \mathcal{E}_{lin}$$

Therefore, we conclude that $\mathcal{E}_{net} \leq c_K$.

$\square$

# B   Architectures in details

In this section we elaborate on the specific architectures that were used in our experiments in Sec. 4. As discussed in Sec. 3.3, all our network architectures are comprised of an opening layer ($1 \times 1$ convolution), a sequence of PDE-GCN layers, and a closing layer ($1 \times 1$ convolution), and possibly additional final convolution steps which serve as the classifier. In total, we have three types of architectures in our experiments, which differ in their classifier layers. Throughout the following tables, $c_{in}$ and $c_{out}$ denote the input and output channels, respectively, and $c$ denotes the number of features in hidden layers (which is a hyper-parameter, as given in Appendix C.) We denote the number of PDE-GCN blocks by $L$, and the dropout probability by $p$.

Our first architecture is described in Tab. 8 and includes only a closing layer as a final step. The architecture is used for the semi- and fully supervised node classification tasks (i.e., the experiment on Cora in Sec. 4.1 – 4.2, the experiments in Sec. 4.3 – 4.4 and the ablation study in Sec. 4.7), as well as the inductive learning task on PPI in Sec. 4.5. Note, the high-level architecture is the same as in GCNII [19], and only differs in the employed GCN-block, which is our PDE-GCN.

Table 8: The architecture used for semi-and fully supervised node classification and inductive learning.

| Input size | Layer | Output size |
|---|---|---|
| $n \times c_{in}$ | $1 \times 1$ Dropout(p) | $n \times c_{in}$ |
| $n \times c_{in}$ | $1 \times 1$ Convolution | $n \times c$ |
| $n \times c$ | ReLU | $n \times c$ |
| $n \times c$ | $L\times$ PDE-GCN block | $n \times c$ |
| $n \times c$ | $1 \times 1$ Dropout(p) | $n \times c$ |
| $n \times c$ | $1 \times 1$ Convolution | $n \times c_{out}$ |

The second architecture is described in Tab. 9, and is used for the ModelNet-10 in Sec. 4.1. The difference between this architecture and the one presented in Tab. 8 is that here we perform a global-max pooling operation to obtain a global shape class prediction. Following this pooling operation, we add two multi-layer perceptron (MLP) layers, where each consists of a $1 \times 1$ convolution, $ReLU$ activation, batch normalization and dropout with probability of $0.5$. Finally, a fully connected convolution layer is applied to obtain the prediction.

Table 9: The architecture used for shape classification on ModelNet-10.

| Input size | Layer | Output size |
|---|---|---|
| $n \times 3$ | $1 \times 1$ Convolution | $n \times c$ |
| $n \times c$ | ReLU | $n \times c$ |
| $n \times c$ | $L\times$ PDE-GCN block | $n \times c$ |
| $n \times c$ | $1 \times 1$ Convolution | $n \times c$ |
| $n \times c$ | ReLU | $n \times c$ |
| $n \times c$ | Global Max-Pool | $1 \times c$ |
| $1 \times c$ | $1 \times 1$ Convolution | $1 \times 128$ |
| $1 \times 128$ | Batch-Normalization | $1 \times 128$ |
| $1 \times 128$ | ReLU | $1 \times 128$ |
| $1 \times 128$ | $1 \times 1$ Dropout(0.5) | $1 \times 128$ |
| $1 \times 128$ | $1 \times 1$ Convolution | $1 \times 64$ |
| $1 \times 64$ | Batch-Normalization | $1 \times 64$ |
| $1 \times 64$ | ReLU | $1 \times 64$ |
| $1 \times 64$ | $1 \times 1$ Dropout(0.5) | $1 \times 64$ |
| $1 \times 64$ | Fully-Connected | $1 \times 10$ |

The third architecture is used for the dense-shape correspondence task on FAUST in Sec. 4.6 is given in Tab. 10. In addition to the closing $1 \times 1$ convolution layer, it also includes a layer of a $1 \times 1$

convolution and an $ELU$ activation, followed by another final $1 \times 1$ convolution which classifies the point-to-point correspondence. In the case of the FAUST dataset, each mesh has $n = 6890$ vertices.

Table 10: The architecture used for dense-shape correspondence on FAUST.

| Input size | Layer | Output size |
|---|---|---|
| $n \times 4$ | $1 \times 1$ Convolution | $n \times c$ |
| $n \times c$ | ReLU | $n \times c$ |
| $n \times c$ | $L \times$ PDE-GCN block | $n \times c$ |
| $n \times c$ | $1 \times 1$ Convolution | $n \times c$ |
| $n \times c$ | ReLU | $n \times c$ |
| $n \times c$ | $1 \times 1$ Convolution | $n \times 512$ |
| $n \times 512$ | ELU | $n \times 512$ |
| $n \times 512$ | Fully-Connected | $n \times n$ |

## C  Hyper-parameters details

We provide the selected hyper-parameters in our experiments, besides for the inductive learning on PPI (Sec. 4.5) and dense shape correspondence (Sec. 4.6) which are reported in the main paper. We denote the learning rate of our PDE-GCN layers by by $LR_{GCN}$, and the learning rate of the $1 \times 1$ opening and closing as well as any additional classifier layers by $LR_{oc}$. Also, the weight decay for the opening and closing layers is denoted by $WD_{oc}$. For the PDE-GCN layers, no weight decay is used throughout all experiments.

### C.1  GCN generalization

For semi-supervised node-classification on Cora, for GCNII we used the same settings as in the original paper of GCNII. For DGCNN and our PDE-GCN$_H$ we used the same hyper-parameters as reported in Tab. 12.

For the ModelNet-10 classification we used a learning rate of $0.01$ without weight decay, for all parameters, on all considered networks, and a hidden feature space of size $c = 64$.

### C.2  Learning PDE dynamics

In this experiment we used a 8 layers mixed PDE-GCN$_M$, starting with $\alpha = 0.5$, such that it is balanced between a PDE-GCN$_D$ and a PDE-GCN$_H$. We report the hyper-parameters for this experiment in Tab. 11.

Table 11: Learning PDE dynamics hyper-parameters

| Dataset | $LR_{GCN}$ | $LR_{oc}$ | $LR_\alpha$ | $WD_{oc}$ | $\#Channels$ | $Dropout$ | $h$ |
|---|---|---|---|---|---|---|---|
| Cora | $1 \cdot 10^{-4}$ | $0.01$ | $0.01$ | $5 \cdot 10^{-4}$ | $64$ | $0.6$ | $0.5$ |
| FAUST | $0.001$ | $0.01$ | $0.01$ | $0$ | $256$ | $0$ | $0.01$ |

### C.3  Semi-supervised node-classification

The hyper-parameters for this experiment are summarized in Tab. 12.

Table 12: Semi-Supervised classification hyper-parameters

| Dataset | $LR_{GCN}$ | $LR_{oc}$ | $WD_{oc}$ | $\#Channels$ | $Dropout$ | $h$ |
|---------|-----------|-----------|-----------|-------------|-----------|-----|
| Cora | $5 \cdot 10^{-5}$ | 0.07 | $5 \cdot 10^{-4}$ | 64 | 0.6 | 0.9 |
| CiteSeer | $2 \cdot 10^{-6}$ | 0.07 | 0.003 | 256 | 0.7 | 0.35 |
| PubMed | $3 \cdot 10^{-5}$ | 0.03 | $1 \cdot 10^{-4}$ | 256 | 0.7 | 0.7 |

## C.4 Fully-supervised node-classification

The hyper-parameters for this experiment are summarized in Tab. 13.

Table 13: Fully-Supervised classification hyper-parameters

| Dataset | $LR_{GCN}$ | $LR_{oc}$ | $WD_{oc}$ | $\#Channels$ | $Dropout$ | $h$ |
|---------|-----------|-----------|-----------|-------------|-----------|-----|
| Cora | $4 \cdot 10^{-5}$ | 0.06 | $1 \cdot 10^{-4}$ | 64 | 0.6 | 0.65 |
| CiteSeer | $2 \cdot 10^{-4}$ | 0.07 | $1 \cdot 10^{-4}$ | 64 | 0.6 | 0.4 |
| PubMed | $5 \cdot 10^{-5}$ | 0.02 | $3 \cdot 10^{-4}$ | 64 | 0.5 | 0.55 |
| Chameleon | $40 \cdot 10^{-4}$ | 0.02 | $8 \cdot 10^{-5}$ | 64 | 0.6 | 0.55 |
| Cornell | $2.5 \cdot 10^{-4}$ | 0.07 | $2.5 \cdot 10^{-4}$ | 64 | 0.5 | 0.05 |
| Texas | $3 \cdot 10^{-4}$ | 0.05 | $1 \cdot 10^{-4}$ | 64 | 0.5 | 0.05 |
| Wisconsin | $3 \cdot 10^{-5}$ | 0.07 | $5 \cdot 10^{-5}$ | 64 | 0.5 | 0.054 |

## C.5 Ablation study

In this experiment we used the same hyper-parameters as reported in Tab. 12.