# OpenReview forum: "PDE-GCN: Novel Architectures for Graph Neural Networks Motivated by Partial Differential Equations"
_NeurIPS.cc/2021/Conference — NeurIPS 2021 Poster_

### Official Review · Reviewer_zFMQ · 2021-07-16

**Rating:** 6
**Confidence:** 3

**Summary:**

The authors propose a new class of graph neural networks motivated by numerical methods for solving PDEs on manifolds. The idea is that there is a correspondence between specific GNNs (e.g., GNNs that learn the weight of edges such as GATs) and discretized PDEs on manifolds. Here, the discretization is spatially. For time (i.e., the depth of the GNN) one can use numerical solvers for PDEs similar to what was proposed by the original Neural ODE paper. The main motivation is that there are types of PDEs that do not suffer from the oversmoothing problem and, therefore, that their discretized versions might be a better fit for some problem domains where oversmoothing is a problem.

**Limitations And Societal Impact:**

The authors claim that the limitations of the paper are sufficiently discussed since "Our theorems state that the activation function must be sign-preserving, bounded and monotone."

What I am wondering about: what about efficiency of the method? What about the dependence on defining W the way you did?

**Main Review:**

First of all I want to say that the idea of drawing connections between PDEs/ODEs and GNNs as spatially discretized PDEs is really interesting. I also think that the experiments are comprehensive and while the improvements are marginal at best, they still validate the idea of relating PDEs on manifolds and GNNs as one that can lead to interesting new architectures.

Unfortunately, I have some concerns when it comes to positioning the work in the context of existing methods. There are several papers that should be discussed here:

[1] Li, Z., Kovachki, N., Azizzadenesheli, K., Liu, B., Bhattacharya, K., Stuart, A., and Anandkumar, A. Multipole graph neural operator for parametric partial differential equations. In NeurIPS, 2020b.

[2] Belbute-Peres, F. d. A., Economon, T., and Kolter, Z. Combining differentiable pde solvers and graph neural networks for fluid flow prediction. In ICML. PMLR, 2020

I would ask the authors to discuss the differences in their rebuttal. Especially the latter seems highly related but perhaps more focused on a particular dynamical system? Moreover, there is an ICML 2021 paper that showed up on arxiv about a month ago:

[3] https://arxiv.org/abs/2106.10934

While [3] appeared on arxiv *after* the NeurIPS submission deadline and therefore, is  not a reason to reject the current submission!, it would still be beneficial to write a few sentences outlining the differences and similarities. One difference I see is the way the edge weight matrix is modelled (in the submission denoted by W and modelled as the geometric mean of the degree of adjacent nodes, in [3] denoted by A and modelled as a GAT). There are also differences in the numerical solvers used.
[3] also compares to several other *ODE* based graph neural networks which, at least, should be discussed in the related work section of the submission.

Again, I don’t think there is a problem with several papers coming out at roughly the same time with roughly the same ideas, as it often points to an important topic ripe for dissemination. But please make sure you end of discussing prior work (even if only motivated by ODEs) on combining differential equations in some form with GNNs.

Besides the related work section, the one area where I see a lot of room for improvement is the presentation. For instance, the order in which you introduce parts of the paper seems odd and counterintuitive. For instance, you state two theorems on the third page. At this point, however, they seem out of context and, in their statement, use terms such as PDE-GCN that have not been introduced before. Either you state these theorems as standalone background theory about PDEs on manifolds, or you first introduce the concepts needed to understand their premises.


**Time Spent Reviewing:**

1 hour

---

> ### Author Response · Authors · 2021-08-10
> **Response to reviewer zFMQ**
>
> We thank the reviewer for the positive assessment of our manuscript and the detailed comments.
>
> Re missing literature Belbute-Peres et al. ICML, 2020 and Li et al. Neurips 2020: While both these papers and ours consider graph neural networks and PDEs, the usage and purpose are different.
> Belbute-Peres et al. utilize GCNs for enhancing the solution of classical partial differential equations (like fluid flow dynamics). There, classical PDE discretization methods are used for the CFD solution. The authors augment the solution with the network GCN [3]. Similarly, Li et al. also target solution of PDEs, and use GCNs as a platform to apply the discretizations. In addition, they apply multiscale architectures to enhance the solution of the PDE and to capture long interactions. In our paper we mention approaches of neural networks for PDEs in line 72, references [37,38]. In the revised version we will add the mentioned citations as methods with similar aims that apply graph neural networks.
> In contrast, we define a novel graph convolution and architectures using PDE discretization concepts. We target a wide array of GCN applications, also focusing on the over-smoothing phenomenon. To summarize, these are two different contributions for different applications and purposes. We will add this reference and discussion to the related work section.
>
> Re the method GRAND (ICML 2021): The method GRAND is indeed relevant for our work and targets similar applications. The authors utilize attention mechanism with diffusive dynamics, and harness several integration schemes to apply the network.  GRAND does not offer the hyperbolic variant which is a significant part of our work, as depicted from the geometric experiment in Table 6. We will add the reference to GRAND in the revised version. In addition, regarding the results comparison: considering Table 1 in GRAND, the accuracy for Cora|Citeseer|PubMed of our PDE-GCN-D is 84.3|75.6|80.6 % versus 84.7|73.3|80.4 % for GRAND-l, 83.6|70.8|79.7 % for GRAND-nl, and 82.9|73.6|81.0 % for GRAND-nl-wr. In all cases our method is better than all the GRAND variants on at least 2 out of 3 benchmarks.
>
> Re efficiency and W: As evident from our attached code, our method is implemented in native PyTorch and includes simple binary operations (additions and subtractions) on the graph to obtain the discretized spatial operators, followed by simple conv1D operations. The weighting matrix W is obtained from the standard degree-normalization as in GCN [3] and GCNII [18], which is done as a pre-processing step.
>
> Re presentation: We appreciate your comment. The order of this part will be improved for the camera-ready version, if accepted.
>
>
> We again would like to thank the reviewer for the positive assessment. We believe that our paper is strong and we are confident that we can address the concerns raised by all the reviewers towards the camera-ready version if accepted. We sincerely hope that the reviewer will consider improving the rating based on the responses.

---

> > ### Comment · Reviewer_zFMQ · 2021-08-27
> > **Response to rebuttal**
> >
> > I have read the rebuttal and will keep my score.

---

### Official Review · Reviewer_Buis · 2021-07-16

**Rating:** 6
**Confidence:** 3

**Summary:**

This paper proposed using discretized PDE operator on the graph to generalize current graph convolutions. With diffusion and hyperbolic equations, the proposedd method has the flexibility to fit the unseen graph signals.

**Limitations And Societal Impact:**

- Justify the motivation of the proposed method over the related works, including continuous PDE, ResNet for GCN, etc.
- Many graph data are suitable for diffusion since the signals are in low-frequency. It's better to use proper datasets to show the advantage of the proposed method, such as a mix of high and low-frequency.


**Main Review:**

This paper proposed discretized PDE to generalize graph convolution.

### Motivation
- The used PDE include non-linear diffusion and hyperbolic. However, the justification for using such two equations is not clear. (Why using these two rather than the other first/second-order PDE?)
- several related works are missing, like _Combining differentiable PDE solvers and graph neural networks for fluid flow prediction_, ICML 20 _Continuous Graph Neural Networks_, ICML 20

### Technical analysis
- Some related work generalized discretized graph convolution into continuous operators. This work generalized graph convolution as discretized operators. In theory, continuous operators should be more flexible than being discretized. So this methodology may need to be discussed.
- If the motivation of using diffusion and hyperbolic is to adapt unseen data, especially conserving and/or smoothing the signals. Then many existing works regarding solving over-smoothing have similar logic, such as _DeepGCN_, ICCV 19.

### Experiment
- The node classification didn't show significant and consistent superiority in performance (many baselines take the lead)
- The protein-protein interaction task showed that the proposed method is among the best, but it is still not the best.
- In Table 7, the proposed method seems also has an over-smoothing issue: performance decrease as layers go deeper

**Time Spent Reviewing:**

6

---

> ### Author Response · Authors · 2021-08-10
> **Response to reviewer Buis**
>
> We thank the reviewer for the thoughtful review.
>
> Re the choice of PDEs: The motivation for the choices of PDEs is discussed in lines 98-102, and is illustrated in Figure 1. We also reference [39,48] which are classical methods in the field of computer vision that utilize anisotropic diffusion (similar to  PDE-GCN-D), and [41] which uses properties of the wave equation for the dense shape correspondence task (similar to our PDE-GCN-H). In addition, line 133 motivates the choice of diffusive dynamics following its similarity to the widely used ResNet architecture. Additionally, both the diffusive and hyperbolic dynamics were considered in the context of CNNs [19,20], obtaining inspiring performance in image classification tasks.
>
> Re missing literature Belbute-Peres et al. ICML, 2020: While both this work and ours involve graph neural networks and PDEs, the usage and purpose are different. Belbute-Peres et al. utilize GCNs for enhancing the solution of classical partial differential equations (like fluid flow dynamics). There, classical PDE discretization methods are used for the CFD solution. The authors augment the solution with the network GCN [3]. In contrast, we define a novel graph convolution and architectures using PDE discretization concepts. We target a wide array of GCN applications, also focusing on the over-smoothing phenomenon. To summarize, these are two different contributions for different applications and purposes. We will add this reference and discussion to the related work section.
>
> Re comparison with continuous GCNs: Continuous graph convolution operators are indeed a powerful tool, since they can be discretized using various approaches. However, in practice every operator is discretized.
> There are two known approaches for this procedure. The first is "Optimize and Discretize" - first define the derivatives and optimality conditions, then discretize. The other, is to "Discretize and Optimize", where a particular discretization is chosen and then the ODE/PDE is discretely solved. We have chosen the latter, similar to many applications in fluid flow, electromagnetics, optimal control and others. The discussion of which approach to adopt is has been well studied both for classical methods (see Perspectives in Flow Control and Optimization by Max Gunzburger), and also recently in the deep-learing community (see Neural Ordinary Differential Equations [Chen et al.] and ANODEV2: A Coupled Neural ODE Framework [Zhang et al.]). The references to those sources and discussion will be added to the revised manuscript. In any case, both options can be used, and it's a matter of taste.
>
> Re DeepGCN: We thank the reviewer for the DeepGCN reference, which we will add to the revised manuscript. This work suggests to utilize dense connections in GCNs, i.e., like in the popular DenseNet CNN. Such dense connections introduce ``highways'' and shorten distances between the different layers and the output of the network. Indeed, this can mitigate the over-smooting phenomenon (similar to the addition of the input features in GCNII). But, dense connections have some computational overhead and therefore DenseNets are usually less wide than, say, ResNets because of those dense connections. In addition, DeepGCN focuses on geometrical applications, while we show generalization for geometrical and non-geometrical applications. In particular, DeepGCN considers different data-sets than ours, hence the direct comparison between the approaches is not available.
>
> Re experiments: We show in Table 3 that we obtain superior results for Citeseer (by over 2%) and Pubmed, while we are surpassed only by GCNII for Cora (by 1%). In Table 4 we outperform all methods on 6 out of 7 data-sets. Namely, we obtain a healthy margin (of over 10%) on Cornell, Texas and the Wisconsin data-sets. Furthermore, in Table 6 (Dense Shape Correspondence) we outperform all other methods. In our opinion these are objectively strong results.
>
> Re PPI: Our PDE-GCN-D obtained 99.07% vs 99.58% by GCNII*. However, we think that showing a variety of different experiments highlights the robustness and generality of our architectures, as mentioned in lines 9-11 and can also be seen from the experiment in Section 4.1. We believe that the result in Table 5, which is high, contributes to this end.
>
> Re over-smooting in Tab. 7: The purpose of Table 7 is to delve on the different components of our method and the relevance of Theorems 1 and 2 to the numerical experiments. In particular, the experiments for the non-symmetric PDE-GCN on Cora and Citeseer indeed suggest that a symmetric operator as formulated in Equations (6)-(7) is beneficial.
> On the contrary, in all other cases, where a symmetric operator is given, we show that no over-smoothing occurs (we note, that a slight degradation of under 0.5% can appear in some cases, which is also common in other methods such as GCNII, but no clear trend of degradation as more layers are stacked is observed). We will further stress that in our explanation in lines 265-266.
>
> Re high-frequency data: Our experiments involve data-sets and applications from different domains, which are both geometrical and non-geometrical. While it is true that most of the popular data-sets are of low frequency, we also experiment with data like FAUST which includes human meshes in different poses, with large perturbations between them, where the goal is to map each vertex to its corresponding vertex in a different mesh. Such application requires conservation (that is, the geodesics need to be conserved), which implies that high frequencies need to be conserved as well. As discussed in lines 98-102 and lines 254-255, it is indeed that for this kind of data-set, the hyperbolic model reads significantly better results than the diffusive model (99.9% compared to 64.2%).
>
>
> We again would like to thank the reviewer for thoughtful review. We believe that our paper is strong and we are confident that we can address the concerns raised by all the reviewers towards the camera-ready version if accepted. We sincerely hope that the reviewer will consider improving the rating based on the responses.

---

> > ### Comment · Reviewer_Buis · 2021-08-27
> > **Response to rebuttal**
> >
> >
> > I have read the rebuttal and updated my score.

---

### Official Review · Reviewer_Pd3f · 2021-07-18

**Rating:** 6
**Confidence:** 4

**Summary:**

Motivated by the connections between PDEs and conventional convolutional networks, this paper proposes the view of graph convolutional networks (GCNs) as discretizations of PDEs on manifolds. Using this view, the authors propose a a family of graph convolutional architectures that utilize discrete graph operators as their basis and argue that this approach can address existing issues with graph convolutional networks, such as over-smoothing. Moreover, the authors also argue that this approach can have a more even performance accorss a diverse range of tasks, instead of being specialized to only a specific type of problem.

**Limitations And Societal Impact:**

The authors have adequately addressed the potential negative societal impact of their work.

As the authors state, they do address some limitations, in the form of the conditions for their theorems to apply. Further discussion of limitations (e.g. practical aspects) might have been of interest, but is not a significant detriment to the paper. The empirical demonstration in Section 4.2 that this argument is reflected in practice (when a mixture model is given a choice of which type of dynamics to use for different types of problems) is an interesting connection to the theoretical analysis.


**Main Review:**

The work is presented clearly and the paper is easy to follow and understand.

The authors present an analysis of the two types of dynamics (diffusion and hyperbolic) they propose as a basis for their graph convolutional architecture. They connect this analysis to the issue of over-smoothing and the different performance of architectures on different type of tasks. It is this analysis that forms the basis for the proposed architecture

The empirical evaluations presented in the paper are comprehensive, including diverse relevant baselines. The proposed method does perform well across a variety of tasks, either matching or surpassing existing methods.

Nevertheless, even though the choice is argued as being motivated by the characteristics of the task at hand, the authors do have the advantage point of proposing two methods and having the possibility of choosing between them for the convenience of the results. Since a mixture model was proposed and evaluated on a few tasks, it would be interesting to know if that single "unified" model is able to maintain the performance of the separate models in the cases where one significantly outperforms the other.

As a final remark, it would be interesting if the authors could expand on the comment on line 229 ("We argue that the success of our network stems from its capability of a priori extracting features from graphs, due to its utilization of discretized differential operators and PDE guided construction"). What specifically in the formulation allows it to better extract information a priori? A better discussion, even if not possible to completely ground it in theory of empirical results, would be important when making such a claim.

Overall, despite some of the comments above, the paper does present an interesting analysis and a good empirical evaluation with good results on important datasets when compared to standard existing methods.

-------------------

Update post authors' response:

The reported experiments with the "unified" model strengthen the experimental evaluation, by showing that the results are not a consequence of arbitrary model-task combinations.

Overall, I originally had only small issues with this paper, and these were addressed by the authors. My previous assessment already reflected the evaluation that "the paper does present an interesting analysis and a good empirical evaluation with good results on important datasets". The authors' response confirms my positive evaluation and I will thus maintain my evaluation and score.



**Time Spent Reviewing:**

I do not review my assigned papers sequentially and thus have not estimated how much time was spent on each paper.

---

> ### Author Response · Authors · 2021-08-10
> **Response to reviewer Pd3f**
>
> We thank the reviewer for the positive assessment of our manuscript and the detailed comments.
>
> Re unified model: It is evident from Figure 2 that the unified model favors the diffusive model in the node-classification task, and the hyperbolic model is preferred  for the shape correspondence problem, where in both cases it achieves the same accuracy as the corresponding chosen single model. To further clarify this interesting question, we also considered other data-sets.
>     For example the for the inductive learning on PPI (Table 5), the single models PDE-GCN-D and PDE-GCN-H got 99.07% and 83.2% respectively. The unified model obtained 99.18%, surpassing the better option.
>     On the semi-supervised experiments in Table 3, the unified model obtained accuracy which is up to +-0.2% different than the proposed method. Also, following this discussion we applied the unified model for the experiments in Table 4 where it consistently surpassed the better reported model (which was already state-of-the-art for 6 out of 7 of the considered data-sets) by 0.35% on average. We will add those results to the revised version.
>
> Re prior from PDE construction of our network: It is stated in lines 148-149 that the weights matrix K in Equations (6),(7),(9) is initialized as an identity matrix, which results in a standard, well-defined PDE. This initialization serves as a prior of the different networks to behave similarly to classical methods [39,41,48] which utilize the diffusive/hyperbolic PDEs for similar imaging and shape analysis tasks. We will further clarify this in the revised paper.
>
> We again would like to thank the reviewer for the positive assessment. We believe that our paper is strong and we are confident that we can address the concerns raised by all the reviewers towards the camera-ready version if accepted. We sincerely hope that the reviewer will consider improving the rating based on the responses.

---

> > ### Comment · Reviewer_Pd3f · 2021-08-30
> > **Response**
> >
> > Thanks for the response. The results with the "unified" model do indeed strengthen the experimental evaluation, by showing that the results are not a consequence of arbitrary model-task combinations.
> >
> > Overall, my previous assessment already reflected the evaluation that "the paper does present an interesting analysis and a good empirical evaluation with good results on important datasets when compared to standard existing methods". I will thus maintain my positive evaluation and score.

---

> ### Author Response · Authors · 2021-08-30
> **We would be happy to answer any further questions**
>
> Thank you for your positive review. We followed your request regarding the unified model, which consistently improved our results throughout the paper, as described in our rebuttal. We hope you are satisfied with the new additional results, and we would be happy to provide more clarifications if needed. We kindly ask you to consider reflecting your positive view of the paper in the score.

---

### Decision · Program_Chairs · 2021-09-27

**Decision:**

Accept (Poster)

**Comment:**

All ratings were (weak) "accept".

Generally the work is novel and timely. A related paper appeared on arxiv in late June (GRAND), which one reviewer pointed to, but obviously this paper was submitted before that appeared, and doesn't factor into this paper's evaluation (as the reviewer appropriately recognized). To me this supports the relevance and importance of this work. My sense was that the reviewers' ratings were low given the feedback they provided, and the authors addressed the reviewers thoroughly and appropriately, causing one to raise their score.

If accepted, my suggestion to the authors is that they do a careful revision of the text to emphasize the broader scope and applicability of this work, in case some of the reviewers' 6 ratings were due to not clearly seeing connections to other areas.